# Chat GPT in Diagnostic Human Pathology: Will It Be Useful to Pathologists? A Preliminary Review with 'Query Session' and Future Perspectives

Gerardo Cazzato [1,*] , Marialessandra Capuzzolo [1], Paola Parente [2] , Francesca Arezzo [3], Vera Loizzi [3] , Enrica Macorano [4] , Andrea Marzullo [1], Gennaro Cormio [3] and Giuseppe Ingravallo [1]

1. Section of Molecular Pathology, Department of Precision and Regenerative Medicine and Ionian Area (DiMePRe-J), University of Bari "Aldo Moro", 70124 Bari, Italy; m.capuzzolo@studenti.uniba.it (M.C.); andrea.marzullo@uniba.it (A.M.); giuseppe.ingravallo@uniba.it (G.I.)
2. Pathology Unit, Fondazione IRCCS Ospedale Casa Sollievo della Sofferenza, 71013 San Giovanni Rotondo, Italy; paolaparente77@gmail.com
3. IRCCS Istituto Tumori "Giovanni Paolo II", 70124 Bari, Italy; francesca.arezzo@uniba.it (F.A.); vera.loizzi@uniba.it (V.L.); gennaro.cormio@uniba.it (G.C.)
4. Section of Legal Medicine, Interdisciplinary Department of Medicine, University of Bari "Aldo Moro", 70124 Bari, Italy; enricamacorano@gmail.com
* Correspondence: gerardo.cazzato@uniba.it; Tel.: +39-340-520-3641

**Abstract:** The advent of Artificial Intelligence (AI) has in just a few years supplied multiple areas of knowledge, including in the medical and scientific fields. An increasing number of AI-based applications have been developed, among which conversational AI has emerged. Regarding the latter, ChatGPT has risen to the headlines, scientific and otherwise, for its distinct propensity to simulate a 'real' discussion with its interlocutor, based on appropriate prompts. Although several clinical studies using ChatGPT have already been published in the literature, very little has yet been written about its potential application in human pathology. We conduct a systematic review following the Preferred Reporting Items for Systematic Reviews and Meta-Analyses (PRISMA) guidelines, using PubMed, Scopus and the Web of Science (WoS) as databases, with the following keywords: ChatGPT OR Chat GPT, in combination with each of the following: pathology, diagnostic pathology, anatomic pathology, before 31 July 2023. A total of 103 records were initially identified in the literature search, of which 19 were duplicates. After screening for eligibility and inclusion criteria, only five publications were ultimately included. The majority of publications were original articles (*n* = 2), followed by a case report (*n* = 1), letter to the editor (*n* = 1) and review (*n* = 1). Furthermore, we performed a 'query session' with ChatGPT regarding pathologies such as pigmented skin lesions, malignant melanoma and variants, Gleason's score of prostate adenocarcinoma, differential diagnosis between germ cell tumors and high grade serous carcinoma of the ovary, pleural mesothelioma and pediatric diffuse midline glioma. Although the premises are exciting and ChatGPT is able to co-advise the pathologist in providing large amounts of scientific data for use in routine microscopic diagnostic practice, there are many limitations (such as data of training, amount of data available, 'hallucination' phenomena) that need to be addressed and resolved, with the caveat that an AI-driven system should always provide support and never a decision-making motive during the histopathological diagnostic process.

**Keywords:** ChatGPT; chatbot; artificial intelligence; AI; pathology; histology

## 1. Introduction

Artificial Intelligence (AI) has revolutionized medical and scientific fields in just a few years, allowing for significant changes and the integration of diagnostic, therapeutic and patient care pathways [1]. Although at first it was mainly represented by the development of Machine Learning (ML) models [2], further advances such as Deep Learning (DL) with,

among others, Convolutional Neural Networks (CNN) soon came to the fore [3]. A branch of AI includes conversational artificial intelligence, which has experienced unprecedented development in recent years, with numerous models and platforms developed to enable machines to understand and respond to natural language input [4]. In more detail, a chatbot is an item of software that simulates and develops human conversations (spoken or written), allowing users to interact with digital devices as though they were speaking with real people [5]. Chatbot might be as basic as a program that responds to a single enquiry or as complex as a digital assistant that learns and develops as it gathers and elaborates information to provide higher levels of personalization [6]. The chatbots designed specifically for activities (declarational) are 'single-purpose' software that focus on carrying out a certain function; regulated responses to user requests are generated using Natural Language Process (NLP) and very little machine learning [7]. The interactions with these chatbots are quite particular and structured, and they are best suited for assistance and service functions like frequently asked and consolidated questions. Common questions can be managed by activity-specific chatbots, such as inquiries about working hours or straightforward transactions that do not involve many variables. Even while they employ NLP in a way that allows users to experiment with it easily, their capabilities are still somewhat limited. These are the most popular chatbots right now [7,8]. Virtual assistants, also known as digital assistants or data-driven predictive (conversational) chatbots, are significantly more advanced, interactive and customized than task-specific chatbots. These chatbots use ML, NLP and context awareness to learn. They employ data analysis and predictive intelligence to offer customization based on user profiles and past user behavior. Digital assistants can gradually learn a user's preferences, make suggestions and even foresee needs. They can start talks in addition to monitoring data and rules. Predictive chatbots that focus on the needs of the user and are data-driven include Apple's Siri and Amazon's Alexa [9].

A clear example of such an approach is ChatGPT, an acronym for Generative Pretrained Transformer, which is a powerful and versatile NLP tool that uses advanced machine learning algorithms to generate human-like responses within a conversation (https://chat.openai.com, accessed on 1 July 2023). Released on 30 November 2022 by OpenAI, ChatGPT (version 3.5) was trained until the end of 2021 on more than 300 billion words, with the ability to respond to a huge variety of topics and with the ability to learn from its human interlocutor [10]. In the first few months after its official launch, many papers were published in the purely informatic field, but, as the weeks went by, the medical and scientific fields were also interested, with a particular interest in the education, research and simulation of clinical pictures of patients, as well as applications in hygiene and public health, clinical medicine, oncology and surgery [11].

On the other hand, in the literature there is a paucity of information regarding the reliability of ChatGPT in assisting the routine activity of the pathologist [12]; among other papers, a recent manuscript by Schukow C. et al. [12,13] underlined the lack of studies that evaluate this relationship, focusing more on the three fundamental criteria on which a potential use of ChatGPT should be based: (1) a chatbot should have a strong performance; (2) an ideal chatbot should be freely accessible for public use; (3) it should be trained on known and recoverable data.

In this review paper, we will try to summarize the potential use of ChatGPT in pathological anatomy, discuss the fields of application studied so far, perform some 'query sessions' about pathological topics that could help the pathologist and try to outline future perspectives, with particular regard to present limitations.

## 2. Materials and Methods

A systematic review was elaborated following the Preferred Reporting Items for Systematic Reviews and Meta-Analyses (PRISMA) guidelines, using PubMed, Scopus and Web of Science (WoS) databases before 31 July 2023 with the following terms: ChatGPT OR Chat GPT, in combination with each of the following: pathology, diagnostic pathology,

anatomic pathology. Only articles in English were recorded. Review articles, meta-analyses, observational studies, case reports, survey snapshot studies, letters to the editor and comments to the letters were all included. Other potentially relevant articles were identified by manually checking the references of the included literature. The articles all had to meet the following inclusion criteria: (1) covering pathological anatomy topics in light of the use of ChatGPT, with the opportunity to discuss strengths and/or limitations; (2) the articles had to necessarily relate ChatGPT to pathology. Exclusion criteria were articles that talked about ChatGPT in general or relating it to other aspects not pertaining to pathological anatomy.

An independent extraction of articles was performed by two investigators (G.C. and M.C.) according to the inclusion criteria, before 31 July 2023. Disagreement was resolved by discussion between the two review authors (Figure 1).

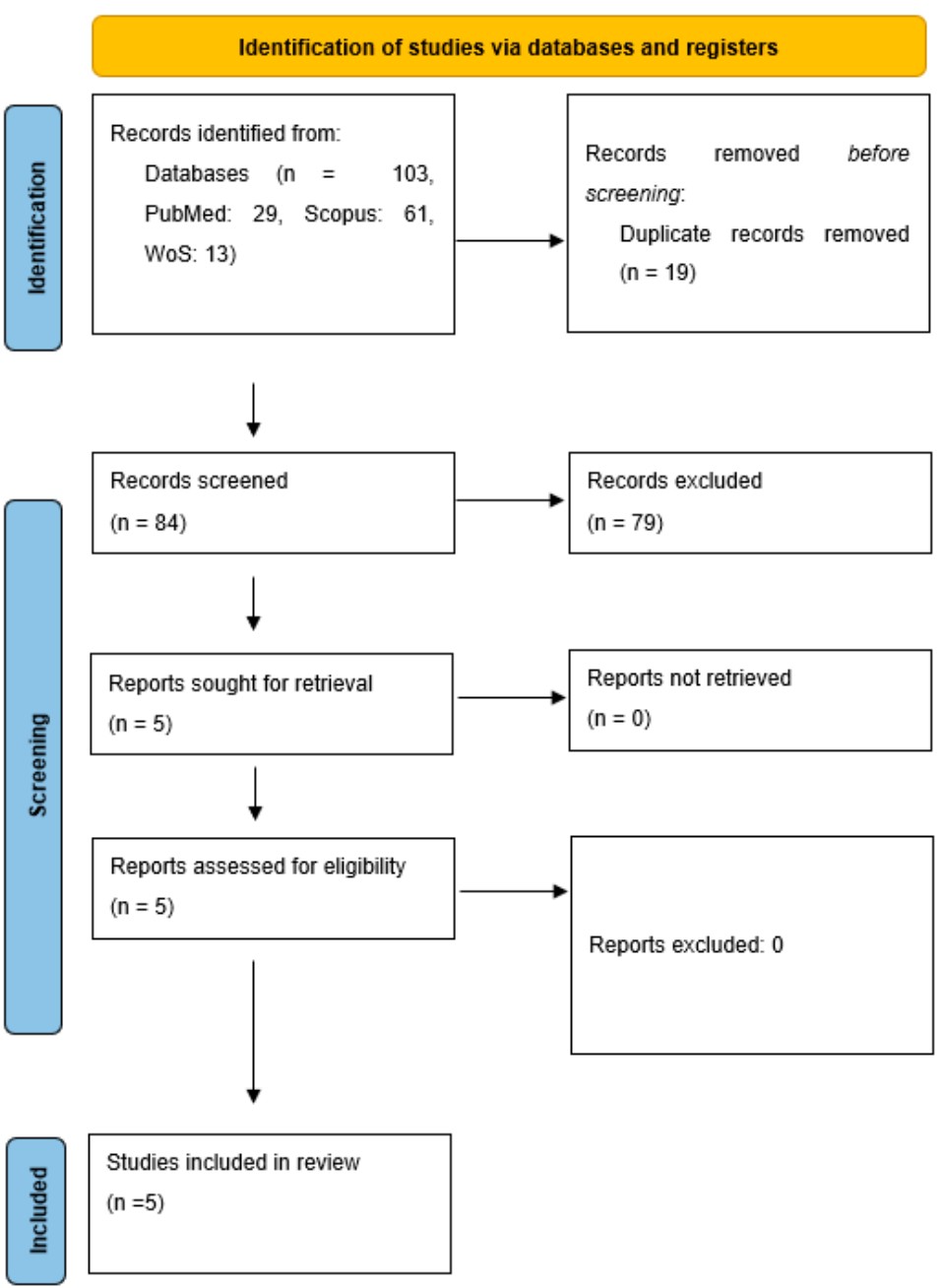

**Figure 1.** PRISMA flow-chart using in this review.

Furthermore, to explore the reliability of ChatGPT in potentially supporting the pathologist's routine diagnostic activity, we developed questions (G.C. and M.C.) that we asked ChatGPT regarding examples of diagnostic approaches in real-life patient cases. Specifically, the first author (G.C.) created a free account on the Open AI site and conducted the conversations with the chatbot. No plugins or other accessories were used in the platform, and all results presented were obtained with ChatGPT version 3.5 and the query sessions were performed between 1 October and 30 October 2023. Some examples of the conversations are summarized in Figures 2–6.

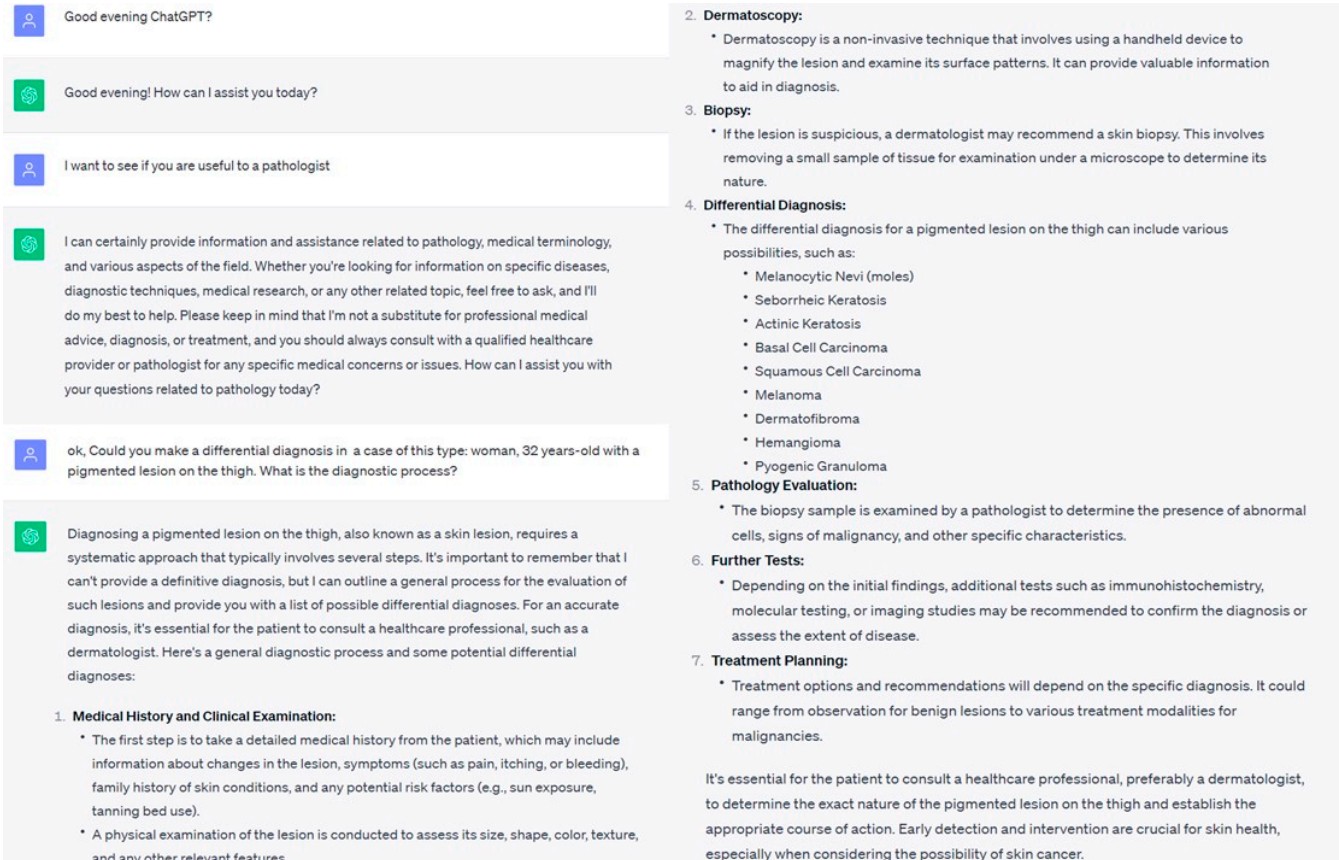

**Figure 2.** The first author (G.C.) posted a request to ChatGPT: "Could you make a differential diagnosis in a case of this type: woman, 32 years-old with a pigmented lesion on the thigh. What is the diagnostic process?" ChatGPT replied, "Diagnosing a pigmented lesion on the thigh, also known as a skin lesion, requires a systematic approach that typically involves several steps. It is important to remember that I cannot provide a definitive diagnosis, but I can outline a general process for the evaluation of such lesions and provide a list of possible differential diagnoses. For an accurate diagnosis, it is essential for the patient to consult a healthcare professional, such as a dermatologist. Here is a general diagnostic process and some potential differential diagnoses [. . .]" (see Figure 2 for the rest). The version of ChatGPT used was dated 25 September 2023, and the average time taken from request entry was about 23.43 s.

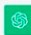

OK, thank you ChatGPT. Now, could you explain to me the clinical criteria for suspecting malignant melanoma?

Certainly, I can provide you with an overview of the clinical criteria for suspecting malignant melanoma, but please keep in mind that I am not a medical professional, and you should consult with a healthcare professional for personalized advice. The diagnosis and evaluation of skin lesions should be done by a qualified dermatologist or healthcare provider.

Malignant melanoma is a type of skin cancer that arises from melanocytes, the pigment-producing cells in the skin. Detecting melanoma involves assessing various clinical features, often using the ABCDE criteria:

1. **Asymmetry:** One half of the mole does not match the other half in terms of shape, size, or color.
2. **Border irregularity:** The edges of the mole are not well-defined; they may be uneven, notched, or blurred.
3. **Color variations:** Melanomas often have uneven coloring, with different shades of brown, black, or even red, white, or blue.
4. **Diameter:** Melanomas are typically larger than common moles. While size alone is not a definitive indicator, moles larger than 6 millimeters (about the size of a pencil eraser) should be examined.
5. **Evolution or change in appearance:** Any change in size, shape, color, or elevation, or the development of new symptoms such as bleeding or itching, should be carefully evaluated.

In addition to the ABCDE criteria, the "EFG" rule is sometimes used:

1. **Evolving:** The mole is changing in size, shape, or color.
2. **Firm:** The mole feels hard or lumpy.
3. **Growing:** The mole is increasing in size.

It's important to note that while these criteria are helpful, not all melanomas exhibit all these characteristics, and some benign moles may exhibit similar features. Regular skin self-exams and professional skin checks are crucial for early detection.

If you have concerns about a mole or skin lesion, it's essential to consult with a healthcare professional, preferably a dermatologist, who can perform a thorough examination and, if necessary, order additional tests such as a biopsy for a definitive diagnosis. Early detection and treatment of melanoma are crucial for a better prognosis.

**Figure 3.** The first author (G.C.) followed up the initial enquiry with another question concerning clinical features useful for suspecting malignant melanoma. ChatGPT provided the ABCDE criteria together with other parameters (detectability, hardness/compactness and lesion growth). The version used was always that of 25 September 2023. ChatGPT had an average response time of 44.32 s.

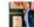

OK, thank you very much ChatGPT. Now could you please explain to me the histolog-ical features of Malignant Melanoma?"

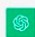

Certainly. Malignant melanoma is diagnosed through a biopsy, where a small sample of the suspicious skin lesion is removed and examined under a microscope. The histological features, or the characteristics observed in the tissue at a microscopic level, play a crucial role in confirming the diagnosis of malignant melanoma. Here are some key histological features associated with malignant melanoma:

1. **Nesting and Invasion:** Malignant melanoma often exhibits a nested or "nest within a nest" growth pattern. Melanoma cells can invade surrounding tissues, including the epidermis (the outer layer of the skin) and the dermis (the deeper layer of the skin).
2. **Asymmetry of Melanocytes:** The melanocytes in melanoma are often irregularly distributed and asymmetrical in shape and size. This is in contrast to normal skin, where melanocytes are more evenly distributed.
3. **Cytological Atypia:** Melanoma cells typically show significant variation in size and shape, a feature known as cytological atypia. This atypical appearance is one of the hallmarks of
4. **Mitotic Figures:** Mitotic figures, which are indicators of cell division, are often increased in melanoma. The presence of a higher number of mitotic figures suggests a more aggressive and rapidly growing tumor.
5. **Pigment Variation:** Melanomas can exhibit variable pigmentation within the tumor, resulting in areas of dark and light pigmentation. This can be seen on microscopic examination.
6. **Inflammatory Infiltrate:** Inflammatory cells may be present around and within the tumor, reflecting the body's immune response to the malignant cells.
7. **Ulceration:** In some cases, melanomas may ulcerate, meaning there is a breakdown of the overlying skin. Ulceration is associated with a poorer prognosis.

It's important to note that there are different subtypes of malignant melanoma, each with its own histological characteristics. The subtypes include superficial spreading melanoma, nodular melanoma, lentigo maligna melanoma, and acral lentiginous melanoma, among others.

The histological examination is a critical component of the diagnostic process and provides valuable information about the nature of the melanoma, its aggressiveness, and the appropriate course of treatment. If you have concerns about a suspicious skin lesion, it's crucial to consult with a dermatologist or healthcare provider for a thorough evaluation and biopsy if necessary.

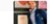

Perfect ChatGPT, and could you explain me "balloon cell melanoma"?

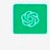

"Balloon cell melanoma" is a rare and distinct variant of malignant melanoma characterized by the presence of balloon cells within the tumor. Balloon cells are large, rounded or oval-shaped cells with clear cytoplasm, giving them a balloon-like appearance under the microscope.

Here are some key features associated with balloon cell melanoma:

1. **Cellular Morphology:** The hallmark feature of balloon cell melanoma is the presence of balloon cells, which are large and have a clear or vacuolated cytoplasm. These cells can sometimes be mistaken for other cell types due to their distinctive appearance.
2. **Melanocytic Origin:** Despite the unusual morphology, balloon cell melanoma is still considered a type of malignant melanoma. The tumor arises from melanocytes, the pigment-producing cells in the skin.
3. **Histological Examination:** Diagnosis is confirmed through histological examination of a biopsy specimen. The biopsy allows pathologists to analyze the cellular characteristics,
4. **Clinical Presentation:** Balloon cell melanoma may present as a pigmented lesion on the skin, similar to other forms of melanoma. It can occur in sun-exposed areas or less commonly in areas not typically exposed to sunlight.
5. **Immunohistochemistry:** Special staining techniques, such as immunohistochemistry, may be used to further characterize the tumor cells and confirm their melanocytic origin. Immunohistochemistry involves using antibodies to detect specific proteins, helping to identify the type of cells present in the tissue.

**Figure 4.** Other answers of ChatGPT to questions proposed by authors.

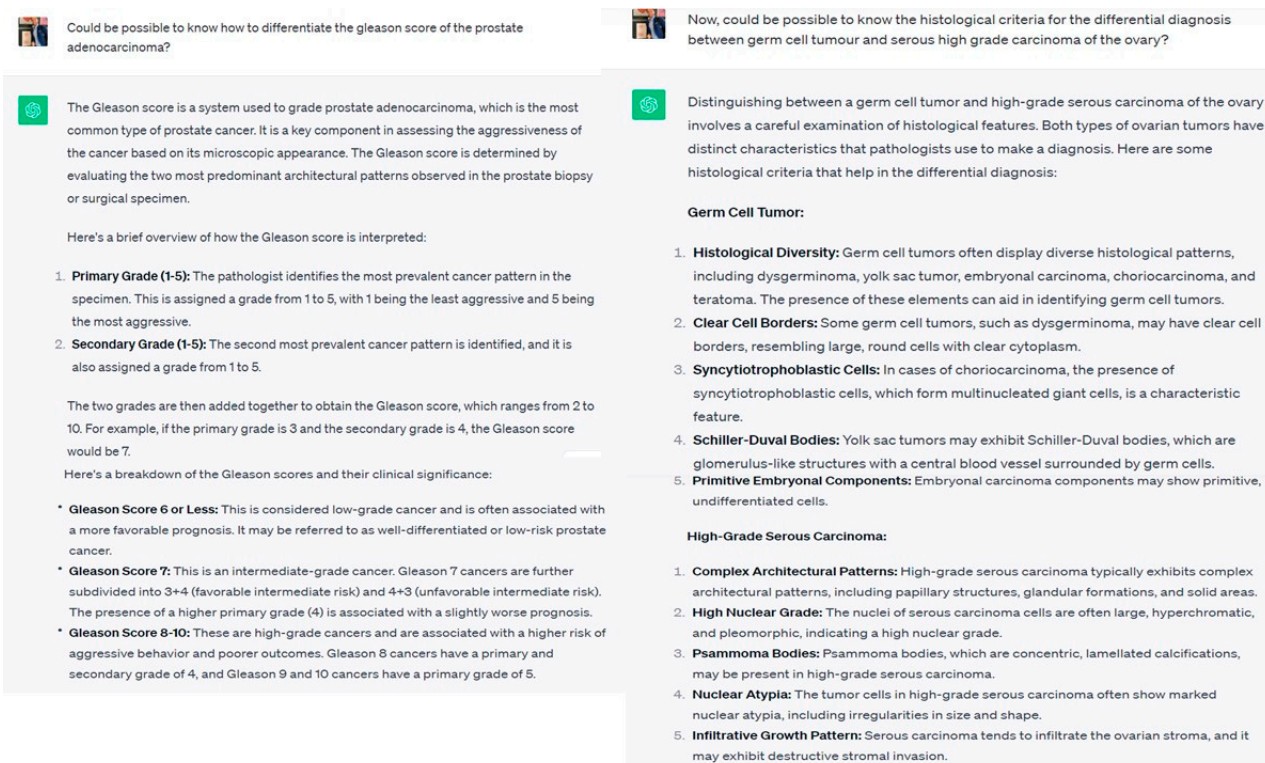

**Figure 5.** Other query sessions in which the first author (G.C.) ask to ChatGPT "..how to differentiate the gleason score of the prostate adenocarcinoma" and "..to know the histological criteria for the differential diagnosis between germ cell tumour and serous high grade carcinoma of the ovary".

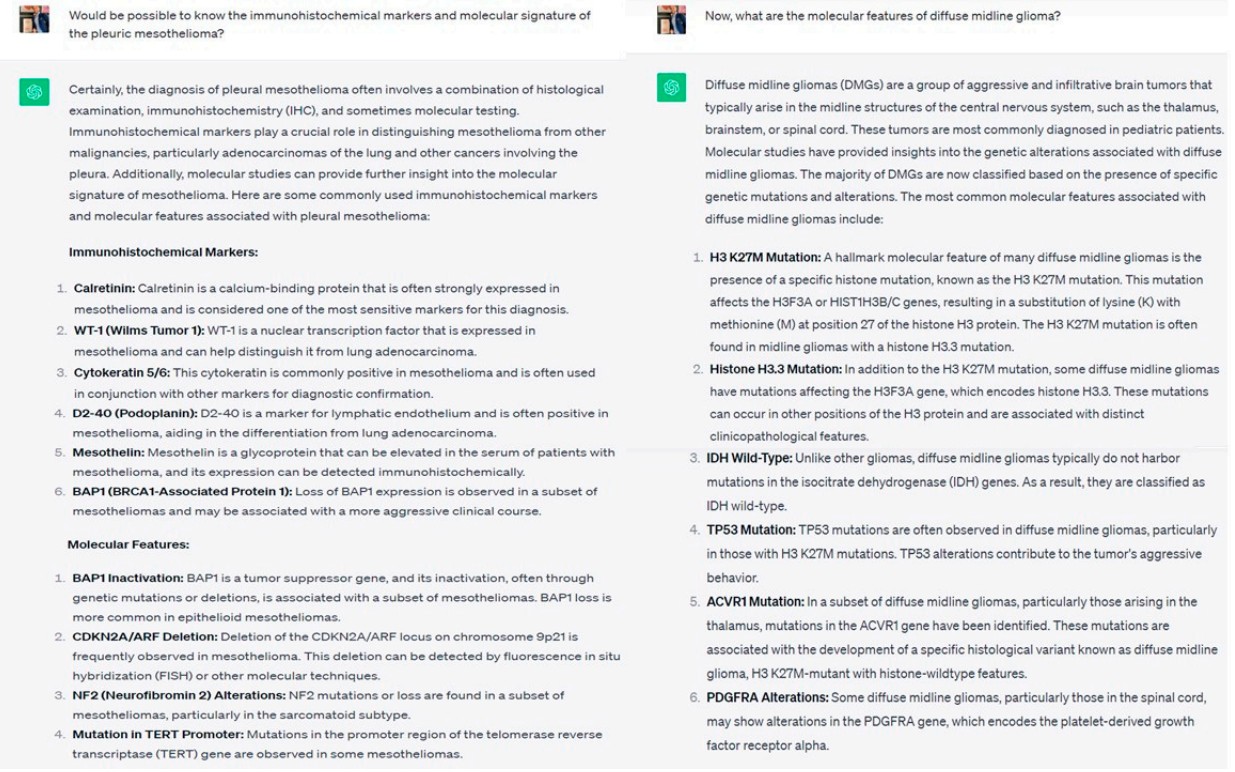

**Figure 6.** Another query session related to immunohistochemical features and molecular characteristics of pleural mesothelioma and to molecular features of pediatric diffuse midline glioma (diffuse intrinsic pontine glioma).

### 3. Results

A total of 103 records were initially identified in the literature search, of which 19 were duplicates. After screening for eligibility and inclusion criteria, only five publications were ultimately included (Figure 1). The majority of publications were original articles (*n* = 2), followed by a case report (*n* = 1), letter to the editor (*n* = 1) and review (*n* = 1).

Figure 1 summarizes the review process following PRISMA guidelines and Table 1 summarizes the features of the five studies included in our review.

**Table 1.** Features (name of author, year, number of reference, type of application of ChatGPT, strengths and weaknesses) of ChatGPT in the publication analyzed in our literature review.

| Year Authors | Type of Paper | Application of ChatGPT | Strengths | Weakness |
|---|---|---|---|---|
| Sallam 2023 [12] | Review | Scientific research | Speeding of the review | Erroneous contents |
| | | | Computer code generation | Hallucination phenomena |
| | | Medical practice | Simplification of the workflow | Risk of incorrect/inaccurate information |
| | | | Improved diagnostics, cost savings, improved health literacy | Transparency and legal issues |
| | | | | Limited knowledge before 2021 |
| | | Health education | | Risk of spreading misinformation |
| | | | | Copyright issues |
| | | | | Lack of originality |
| Sinha [14] 2023 | Article | Query session of 100 questions | Reasonable level of rationality | Lack of true understanding of the underlying significance and context of the information |
| Sorin [15] 2023 | Article | ChatGPT in a molecular tumor board | Clinical recommendations of ChatGPT in line with those of the oncology committee in 70% of the cases | High difficulty in providing empirical decisions on the therapeutic path |
| Naik [16] 2023 | Case Report | ChatGPT in the setting of clinical management | Provide clinical and pathological information | Allucination phenomena |
| Yamin MA [17] 2023 | Article | ChatGPT in gastrointestinal pathology | Summarize patient records | Risk of inaccurate information |
| | | | Incorporation into digital pathology | Allucination phenomena |
| | | | Education and research | |

Regarding the 'query session', the first question was related to the diagnostical and therapeutical approach in a clinical case of a '32 years-old woman with a pigmented lesion on the thigh'. ChatGPT was able to answer correctly and comprehensively, summarizing the different anamnestic stages peculiar to the case, with a satisfactory explanation of concepts such as dermoscopy, biopsy and main differential diagnosis. Furthermore, ChatGPT cautioned against using this information without proper vetting and validation by an experienced pathologist. We consider this first step very satisfactory (ChatGPT's responses were assessed individually by three reviewers who checked each response to the standardized responses to evaluate the response's consistency. Based on their ranking, responses were classified as either "consistent" or "inconsistent." Throughout the evaluation process, there

was strong agreement amongst the three reviewers, guaranteeing a consistent and unbiased assessment of the responses).

Going into more detail, the first author went on to ask ChatGPT: "OK, thank you ChatGPT. Now, could you please explain to me the clinical criteria for suspecting malignant melanoma?" and ChatGPT, in a very performant and correct manner, replied: "Certainly, I can provide you with an overview of the clinical criteria for suspecting malignant melanoma, but please keep in mind that I am not a medical professional, and you should consult with a healthcare professional for personalized advice. The diagnosis and evaluation of skin lesions should be done by a qualified dermatologist or healthcare provider.

Malignant melanoma is a type of skin cancer that arises from melanocytes, the pigment-producing cells in the skin. Detecting melanoma involves assessing various clinical features, often using the ABCDE criteria:

**Asymmetry:** One half of the mole does not match the other half in terms of shape, size, or color.

**Border irregularity:** The edges of the mole are not well-defined; they may be uneven, notched, or blurred.

**Color variations:** Melanomas often have uneven coloring, with different shades of brown, black, or even red, white, or blue.

**Diameter:** Melanomas are typically larger than common moles. While size alone is not a definitive indicator, moles larger than 6 millimeters (about the size of a pencil eraser) should be examined.

**Evolution or change in appearance**: Any change in size, shape, color, or elevation, or the development of new symptoms such as bleeding or itching, should be carefully evaluated.

In addition to the ABCDE criteria, the "EFG" rule is sometimes used:

**Evolving:** The mole is changing in size, shape, or color.

**Firm:** The mole feels hard or lumpy.

**Growing:** The mole is increasing in size.

It's important to note that while these criteria are helpful, not all melanomas exhibit all these characteristics, and some benign moles may exhibit similar features. Regular skin self-exams and professional skin checks are crucial for early detection". (see Figure 3 for the rest of ChatGPT's response).

Again, ChatGPT proved to be quite performant and was able to provide correct answers to the appropriate questions (Figure 3).

Finally, the first author asked ChatGPT: "OK, thank you very much ChatGPT. Now could you please explain to me the histological features of Malignant Melanoma?" and ChatGPT replied: "Certainly. Malignant melanoma is diagnosed through a biopsy, where a small sample of the suspicious skin lesion is removed and examined under a microscope. The histological features, or the characteristics observed in the tissue at a microscopic level, play a crucial role in confirming the diagnosis of malignant melanoma. Here are some key histological features associated with malignant melanoma:

**Nesting and Invasion:** Malignant melanoma often exhibits a nested or "nest within a nest" growth pattern. Melanoma cells can invade surrounding tissues, including the epidermis (the outer layer of the skin) and the dermis (the deeper layer of the skin).

**Asymmetry of Melanocytes:** The melanocytes in melanoma are often irregularly distributed and asymmetrical in shape and size. This is in contrast to normal skin, where melanocytes are more evenly distributed.

**Cytological Atypia:** Melanoma cells typically show significant variation in size and shape, a feature known as cytological atypia. This atypical appearance is one of the hallmarks of malignant cells.

**Mitotic Figures:** Mitotic figures, which are indicators of cell division, are often increased in melanoma. The presence of a higher number of mitotic figures suggests a more aggressive and rapidly growing tumor.

**Pigment Variation:** Melanomas can exhibit variable pigmentation within the tumor, resulting in areas of dark and light pigmentation. This can be seen on microscopic examination.

**Inflammatory Infiltrate:** Inflammatory cells may be present around and within the tumor, reflecting the body's immune response to the malignant cells.

**Ulceration:** In some cases, melanomas may ulcerate, meaning there is a breakdown of the overlying skin. Ulceration is associated with a poorer prognosis.

It's important to note that there are different subtypes of malignant melanoma, each with its own histological characteristics. The subtypes include superficial spreading melanoma, nodular melanoma, lentigo maligna melanoma, and acral lentiginous melanoma, among others.

The histological examination is a critical component of the diagnostic process and provides valuable information about the nature of the melanoma, its aggressiveness, and the appropriate course of treatment. If you have concerns about a suspicious skin lesion, it's crucial to consult with a dermatologist or healthcare provider for a thorough evaluation and biopsy if necessary". Then, G.C. ask to ChatGPT: "Perfect ChatGPT, and could you explain me "balloon cell melanoma"?

ChatGPT

"Balloon cell melanoma" is a rare and distinct variant of malignant melanoma characterized by the presence of balloon cells within the tumor. Balloon cells are large, rounded or oval-shaped cells with clear cytoplasm, giving them a balloon-like appearance under the microscope.

Here are some key features associated with balloon cell melanoma:

Cellular Morphology: The hallmark feature of balloon cell melanoma is the presence of balloon cells, which are large and have a clear or vacuolated cytoplasm. These cells can sometimes be mistaken for other cell types due to their distinctive appearance.

Melanocytic Origin: Despite the unusual morphology, balloon cell melanoma is still considered a type of malignant melanoma. The tumor arises from melanocytes, the pigment-producing cells in the skin.

Histological Examination: Diagnosis is confirmed through histological examination of a biopsy specimen. The biopsy allows pathologists to analyze the cellular characteristics, including the presence of balloon cells, and determine the malignancy of the lesion.

Clinical Presentation: Balloon cell melanoma may present as a pigmented lesion on the skin, similar to other forms of melanoma. It can occur in sun-exposed areas or less commonly in areas not typically exposed to sunlight.

Immunohistochemistry: Special staining techniques, such as immunohistochemistry, may be used to further characterize the tumor cells and confirm their melanocytic origin. Immunohistochemistry involves using antibodies to detect specific proteins, helping to identify the type of cells present in the tissue (Figure 4).

Further, when ChatGPT was asked to provide references on balloon cell melanoma, two non-existent papers were provided (hallucination phenomenon), confirming what had already been reported by several papers previously.

## 4. Discussion

The application of AI to medicine has significantly co-assisted physicians' therapeutic decision-making processes, not replacing but instead complementing and enhancing the indispensable figure of the human [8]. The advent of ChatGPT has further enabled a breakthrough in Large Language Models (LLM) that enable the simulation of a real conversation on a wide variety of topics, including medical and scientific notions [11].

In a recent paper by Sallam [12], the various advantages and limitations of using ChatGPT in areas such as scientific research, practice and health education were analyzed. In particular, scientific research could really benefit from a useful and powerful tool such as ChatGPT by speeding up the process of literature reviews and computer code generation, allowing the human user to focus more on experimental design [13,18–21]; on the other hand, several authors [13,19,22] have highlighted issues of reliability of the data provided

by ChatGPT with the generation of erroneous and/or inaccurate content, phenomena of 'hallucination' (by which is meant the generation of erroneous content but which can be considered plausible from a scientific point of view [23]) and the bias of the answers provided by ChatGPT, which is a reflection of the dataset used in training [12]. Finally, it is important to consider that ChatGPT may generate nonexistent references, as pointed out by Chen T.J. [24] and Lubowitz [25].

If in the early months the field of application was mostly restricted to clinical medicine, in recent times a number of papers have studied, tested and commented on the applicability of Chat GPT in the field of human pathology, making it possible to outline its real usefulness and current limitations.

The article by Sinha et al. [14] describes a study conducted on ChatGPT's ability to resolve complex rationality problems in the area of human pathology. Based on the clear finding that AI is used to analyze medical images, such as histopathologic slides, in order to identify and diagnose diseases with high precision, the authors take a cautious approach to the fact that NLP algorithms are used to analyze the relationships between pathologies, extract relevant information and aid in disease diagnosis. The goal of the study was to assess ChatGPT's ability to address high-level rational questions in the field of pathology. One hundred questions that were randomly chosen from a bank of inquiries regarding diseases and divided into 11 different systems of pathology were used. Experts have evaluated the responses provided by ChatGPT using both a scale of 0 to 5 and the tassonomy SOLE to assess the depth of understanding demonstrated in the responses.

The outcomes have shown that the responses provided by ChatGPT achieve a reasonable level of rationality, with a score of four or five. This means that AI is capable of correctly responding to high-level inquiries requiring in-depth knowledge of the subject. The report also highlights the limitations of AI in diagnosing diseases. Although it is possible to recognize schemes and categorize data, a true understanding of the underlying significance and context of the information is lacking. AI is unable to make logical judgements or evaluative decisions because it lacks the ability to comprehend personal values and judgements. Therefore, it is suggested that careful consideration should be given to the use of AI in medical education, with the goal of assisting human judgement rather than replacing it.

There are some limitations to the study, including the subjective nature of the evaluation procedure and the selection of particular questions from a single bank of data. The authors suggest that in order to obtain results that are more generally applicable, future studies may be conducted on a larger sample size and by a variety of institutions.

In the paper by Sorin V. et al. [15], the authors discuss how ChatGPT 3.5 can also operate within the molecular tumor board, not only starting from histopathological/diagnostic data, but also integrating other key components such as the genetic or molecular response and/or prediction of treatment response and prognosis data. Ten consecutive cases of women with breast cancer were considered and an attempt was made to assess how consistent the recommendations provided by the chatbot were with those of the tumor board. The results showed that ChatGPT's clinical recommendations were in line with those of the oncology committee in 70% of the cases, with concise clinical case summaries and explained and reasoned conclusions. However, the lowest scores (which were given by the second reviewer) were for the clinical recommendations of the chatbot, suggesting that deciding on clinical treatment from pathological/molecular data is highly challenging, requiring medical understanding and experience in the field. Furthermore, it was curious to note that ChatGPT never mentioned the role of medical radiologists, suggesting that incomplete training (and the consequent risk of bias) may influence the performance, and thus the responses, of the chatbot.

In another recent paper by Naik H.R. et al. [16], the authors described the case of a 58-year-old woman with bilateral synchronous breast cancer (s-BBC) who underwent bilateral mastectomy, sentinel lymph node biopsy (BLS), axillary lymphadenectomy with adjuvant radiotherapy and chemo/hormonotherapy. The particularity of the paper was

related to the so-called 'hallucination phenomenon', i.e., a clear and confident response from ChatGPT but which is not real. One of the authors of the paper (Dr. Gurda), when asking the chatbot about s-BBC, noted that although the answer was plausible, the reference provided did not exist, although there were articles with similar information and authors.

This aspect is well addressed by the paper by Metze K. et al. [26], who conducted a study to assess the ability of ChatGPT to contribute to a review on Chagas disease, focusing on the role of individual researchers. Therefore, 50 names of researchers with at least four publications on Chagas disease were selected from Clarivate's Web of Science (WoS) database and for each researcher, the chatbot was asked to provide conceptual contributions related to the study of the disease. The answers were checked by two observers against the literature and incorrect information was removed. The percentage of correct words in the text generated by ChatGPT was calculated and the literature references were classified into three categories: completely correct, minor errors and major errors.

The results showed that the average percentage of correct words in the text generated by ChatGPT was 59.4% but the variation was wide, ranging from 10.0% to 100.0%. A positive correlation was observed between the percentage of correct words and the number of indexed publications of each author of interest, as well as with the number of citations and the author's H-index. However, the percentage of correct references was very low, averaging 7.07%, and both minor and major errors were found in the references.

In conclusion, the results of this study suggest that ChatGPT is still not a reliable source for literature reviews, especially in more specific areas with a relatively low number of publications, as there are still accuracy and misinformation issues to be addressed, especially in the field of medicine.

Yamin Ma [17], in a paper of July 2023, discusses the application of ChatGPT in the context of gastrointestinal pathology, hypothesizing three possible applications for ChatGPT:

(1)  Ability to summarize patient records: ChatGPT could be integrated into the patent table to summarize patients' previous clinical information, helping pathologists better understand patients' current health status and saving time before case reviews.

(2)  Incorporation into digital pathology: ChatGPT could improve the interpretation of computer-aided diagnosis (CAD) systems in gastrointestinal pathology. It would enable pathologists to ask specific questions on digitized images and obtain knowledge-based answers associated with diagnostic criteria and differential diagnosis.

(3)  Role in education and research: ChatGPT could be used for health education, offering scientific explanations associated with medical terms in pathology. However, attention should be paid to the quality of the training data to avoid biased content and inaccurate information. The use of ChatGPT in research also requires caution as it may be insufficient or misleading.

Finally, the paper emphasizes that while recognizing the potential of ChatGPT, it is important to proceed with caution when using artificial intelligence-based technologies such as ChatGPT in gastrointestinal pathology. The aim should be to integrate such language models in a regulated and appropriate manner, exploiting their advantages to improve the quality of healthcare without replacing human expertise and without ignoring expert consultation in particular cases.

From what has been discussed so far and bearing in mind a paper published a few days ago [13], it would appear that at present, the use of ChatGPT in pathology is still in its early stages. In particular, with regard to ChatGPT version 3.5, it seems clear that the amount of data on which the algorithm has been trained plays a key role in its ability to provide correct answers to certain prompts. In particular, several papers have warned of the risk of possible bias and transparency issues [27,28] and of damage resulting from inaccurate or outright incorrect content [29–32]. One of the most problematic phenomena is hallucination, as ChatGPT seems, at present, to produce correct scientific content but not to direct the content itself to a real source/reference. Therefore, its use in pathology and, more generally, in scientific research must necessarily take these limitations into account [33].

Furthermore, it is important to say that there is need for a new framework on publication/authorship ethics in a new age of AI-sourced digital composition; it is always important to address the hallucination phenomena with a check of the user.

From an ethical point of view, it is very important to understand the issue of patients' private data, and if the use of medical clinical records is necessary, it will be important to find a way of protecting patient information.

*Future Roadmaps*

As highlighted in the work of Schukow et al. [13], it is imperative to outline future perspectives that the implementation of AI models will bring to the fore; first of all, it is important to consider how the use of AI methods applied to the writing of scientific articles will be managed, how to address the issue of consent and whether to modify the editorial lines of journals taking into account the use of chatbots. Secondly, it is very important to understand how ChatGPT can impact the possible option of specialization in pathological anatomy (increase or reduction) and how and to what extent there will be a need for a critical review of AI-generated content and whether or not the role of teacher/mentor can be delegated to ChatGPT.

Projecting to a future in which such systems may become more active, their integration with clinical, genetic, anamnestic, morphological and immunohistochemical data, which have always been key to pathologists' roles, will have to be screened by professionals with medical experience and knowledge, which are areas in which ChatGPT struggles the most.

**Author Contributions:** Conceptualization, G.C. (Gerardo Cazzato) and M.C.; methodology, G.C. (Gerardo Cazzato), F.A. and E.M.; software, G.C. (Gerardo Cazzato), F.A. and V.L.; validation, G.C. (Gerardo Cazzato), M.C., A.M. and G.I.; formal analysis, P.P. and A.M.; investigation, G.C. (Gerardo Cazzato); resources, G.C. (Gerardo Cazzato); data curation, G.C. (Gerardo Cazzato); writing—original draft preparation, G.C. (Gerardo Cazzato); writing—review and editing, G.C. (Gerardo Cazzato) and G.C. (Gennaro Cormio); visualization, G.I.; supervision, P.P., G.C. (Gennaro Cormio) and G.I. All authors have read and agreed to the published version of the manuscript.

**Funding:** This research received no external funding.

**Institutional Review Board Statement:** Not applicable.

**Informed Consent Statement:** Not applicable.

**Data Availability Statement:** Not applicable.

**Conflicts of Interest:** The authors declare no conflict of interest.

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
