# Peer review of "Chat GPT in Diagnostic Human Pathology: Will It Be Useful to Pathologists? A Preliminary Review with ‘Query Session’ and Future Perspectives"

_ai, doi:10.3390/ai4040051_

Round 1

Reviewer 1 Report

Comments and Suggestions for Authors

The article is devoted to a review of the question “Can ChatGPT be actively used by pathologists?” From the point of view of artificial intelligence, the article is of absolutely no interest. From a medical point of view, the conclusions are obvious and predictable even at the stage of reading the title of the article. Naturally, ChatGPT cannot be actively and widely used by pathologists due to a number of limitations imposed by the peculiarities of the training conducted. An area in medicine where ChatGPT can be useful is the analysis of digital images of patient tests.

The article is not very well structured. The central goals of the authors that they set for themselves when writing the article are not clear. The conclusions are too banal and obvious.

A number of corrections should be made to the text of the article, which will significantly strengthen the semantic component of the manuscript.

Author Response

Dear Reviewer n’1, first of all thank you very much for your precious suggestions. We will try to do our best to improve the quality of our manuscript. 

The goal of our review is to highlight the features of LLM as Chat-GPT in the field of Pathology, but, because in literature there are very few papers about this field, we think that is important to study this topic. For there reasons in the Introduction we have underscore the lack of papers about this relationship and, then, we have added a small map of the following review.

we performed a ‘query session’ for ChatGPT regarding some questions about the diagnostic pathology and we added the answers in the figure 2-4. We hope that these features will be sufficient and explanatory.

Reviewer 2 Report

Comments and Suggestions for Authors

The article explores the possible uses of ChatGPT, a conversational AI, in human pathology. It emphasizes its capacity to offer scientific information for regular diagnostic procedures while pointing out challenges that must be addressed. The overall idea is interesting. The main highlighted points follow such as:

1) Introduction 

The introduction delineates the advancements of Artificial Intelligence (AI) in the medical-scientific arena, particularly emphasizing the contributions of conversational AI models like ChatGPT. While it commendably captures the model's ability to simulate human-like conversations and provide personalized interactions, the specific scientific challenges faced by applying such tools in Pathological Anatomy are not lucidly presented.

Readers need to grasp not only the potential advantages of ChatGPT in various medical fields, including education, research, and clinical simulations but also to be cognizant of its limitations and challenges, especially in aiding pathologists. A more precise exposition of the problem statement — specifically, what exact gaps or needs ChatGPT addresses in Pathological Anatomy — would enhance the paper's precision.

Furthermore, concluding the introduction with a brief summary or a roadmap of the subsequent sections of the article might be beneficial, providing readers with a clear navigational guide through your exploration.

In its current form, the introduction provides a broad overview. However, refining it to pinpoint the challenges and providing a structured paper outline would significantly bolster its impact and clarity.

2) Related works and methods

The authors undertook a systematic review by PRISMA guidelines, probing databases including PubMed, Scopus, and Web of Science with keywords connecting ChatGPT to pathology. Of the initial 103 records pinpointed, only 5 met the stringent eligibility and inclusion criteria, making it to the final review stage. These chosen records encompassed a variety of publication types: original research articles, case studies, a letter directed to the editor, and an encompassing review.

The analysis accentuated ChatGPT's capability to furnish vast scientific data pertinent to the routine microscopic diagnosis in human pathology. However, the authors were discerning in highlighting certain constraints, pointing to issues like the quality and quantity of training data, data availability, and occurrences of "hallucination" phenomena. Furthermore, the authors underscored a crucial sentiment: while AI entities like ChatGPT can augment pathologists' expertise, they should refrain from usurping the pivotal decision-making role during histopathological diagnostics.

3) Contributions

The paper comprehensively examines ChatGPT's potential role in Pathological Anatomy and offers insights into its existing applications. Based on the overall idea described in the manuscript, here are some constructive suggestions for the authors:

a) Deep Dive into Specific Applications: While the paper broadly discusses the applications of ChatGPT in Pathological Anatomy, a deeper dive into specific case studies or examples where ChatGPT has been used could offer a tangible understanding to the readers. How was it implemented, and what were the outcomes?

b) Comparative Analysis: Given that only five publications met the review's criteria, a comparative analysis of these studies might offer a richer understanding. How do the findings of these papers align or differ, and what can we infer from these comparisons?

c) Addressing Limitations: The paper astutely points out several limitations, including data-related issues and "hallucination" phenomena. An exploration into potential solutions or workarounds for these limitations could be invaluable. For instance, are there new data sources or training techniques that could help overcome these challenges?

d) Ethical Considerations: Beyond the technical and application-based discussions, it might be beneficial to touch upon the ethical implications of using AI like ChatGPT in a sensitive field like Pathological Anatomy. What are the potential risks, and how can they be mitigated?

e) Future Roadmap: With the foundational knowledge laid out in the paper, readers might benefit from a section discussing the future roadmap or potential evolution of ChatGPT in this domain. What advancements or refinements can we expect in the coming years?

f) Collaborative Frameworks: The emphasis on AI systems being supportive tools rather than decision-makers is crucial. A framework or guideline could be proposed that outlines best practices for pathologists to work collaboratively with AI, ensuring optimal outcomes without compromising the human touch.

Incorporating these suggestions might offer a more holistic perspective on the subject and further enrich the paper's contributions to the field.

4) Results

While the authors have made a noteworthy effort to explore the potential of ChatGPT in the realm of Pathological Anatomy, there are several critical concerns regarding the robustness and comprehensiveness of the results presented:

a) Limited Publications: From the outset, identifying 103 records and narrowing them down to merely five is quite limiting. The drastic reduction raises questions about the breadth of research available on the topic or the perhaps overly stringent inclusion criteria used by the authors. Such a limited pool can introduce biases and reduce the generalizability of the findings.

b) Nature of Included Publications: Most publications were original articles, which, while valuable, represent a narrow slice of the scientific literature. Including only one case report, a letter to the editor, and a review must offer a diverse enough perspective on the potential applications and challenges of using ChatGPT in pathology.

c) Reliance on a 'Query Session': The decision to conduct a singular 'query session' to evaluate ChatGPT's competence may not offer a sufficiently rigorous assessment. Real-world scenarios often entail many complex cases, each requiring distinct considerations. A single session can hardly indicate the AI's capability across a diverse set of pathologies.

d) Absence of Comparative Analysis: The paper does not compare ChatGPT's responses to those of experienced pathologists or other AI systems. Such a comparative approach would offer a more explicit benchmark of the model's efficacy.

e) Lack of Quantitative Metrics: The authors deemed the 'query session' results satisfactory. However, the criteria or metrics that led to this conclusion must be clarified. A more structured, perhaps quantitative, evaluation offered more weight to this assertion.

f) Over-reliance on AI Caution: While it is commendable that ChatGPT cautions against using its information without expert validation, leaning on this cautionary note as a safety net could be misleading. It is essential to assess the tool's reliability critically rather than depend on its in-built warnings.

g) Preliminary Nature of Results: The results, as presented, seem more exploratory than conclusive. Given the transformative potential of AI in medical diagnostics, a more thorough investigation with a larger dataset, diverse case studies, and multi-faceted evaluations would have been more compelling.

In essence, while the paper taps into a promising area of research, its results, in their current form, appear to be in the nascent stages. There is a pressing need for a more rigorous, comprehensive, and diverse evaluation to substantiate the claims made.

Author Response

Reviewer n’2: The article explores the possible uses of ChatGPT, a conversational AI, in human pathology. It emphasizes its capacity to offer scientific information for regular diagnostic procedures while pointing out challenges that must be addressed. The overall idea is interesting. The main highlighted points follow such as:

Answer n’1: Dear Reviewer n’2, first of all thank you very much for your precious suggestions. We will try to do our best to improve the quality of our manuscript.

Reviewer n’2: 1) Introduction

The introduction delineates the advancements of Artificial Intelligence (AI) in the medical-scientific arena, particularly emphasizing the contributions of conversational AI models like ChatGPT. While it commendably captures the model's ability to simulate human-like conversations and provide personalized interactions, the specific scientific challenges faced by applying such tools in Pathological Anatomy are not lucidly presented.

Readers need to grasp not only the potential advantages of ChatGPT in various medical fields, including education, research, and clinical simulations but also to be cognizant of its limitations and challenges, especially in aiding pathologists. A more precise exposition of the problem statement — specifically, what exact gaps or needs ChatGPT addresses in Pathological Anatomy — would enhance the paper's precision.

Furthermore, concluding the introduction with a brief summary or a roadmap of the subsequent sections of the article might be beneficial, providing readers with a clear navigational guide through your exploration.

In its current form, the introduction provides a broad overview. However, refining it to pinpoint the challenges and providing a structured paper outline would significantly bolster its impact and clarity.

Answer n’2: Dear Reviewer n’2, thank you very much for your suggestions. The goal of our review is to highlight the features of LLM as Chat-GPT in the field of Pathology, but, because in literature there are very few papers about this field, we think that is important to study this topic. For there reasons in the Introduction we have underscore the lack of papers about this relationship and, then, we have added a small map of the following review. Thank you very much.

Reviewer n’2: 2) Related works and methods

The authors undertook a systematic review by PRISMA guidelines, probing databases including PubMed, Scopus, and Web of Science with keywords connecting ChatGPT to pathology. Of the initial 103 records pinpointed, only 5 met the stringent eligibility and inclusion criteria, making it to the final review stage. These chosen records encompassed a variety of publication types: original research articles, case studies, a letter directed to the editor, and an encompassing review.

The analysis accentuated ChatGPT's capability to furnish vast scientific data pertinent to the routine microscopic diagnosis in human pathology. However, the authors were discerning in highlighting certain constraints, pointing to issues like the quality and quantity of training data, data availability, and occurrences of "hallucination" phenomena. Furthermore, the authors underscored a crucial sentiment: while AI entities like ChatGPT can augment pathologists' expertise, they should refrain from usurping the pivotal decision-making role during histopathological diagnostics.

Answer n’3: It’s perfect dear Reviewer n’2. These features are the most important to us. Thanks a lot.

Reviewer n’2: a) Deep Dive into Specific Applications: While the paper broadly discusses the applications of ChatGPT in Pathological Anatomy, a deeper dive into specific case studies or examples where ChatGPT has been used could offer a tangible understanding to the readers. How was it implemented, and what were the outcomes?

Answer n’4: Thank you very much dear Reviewer n’2. Ok, we performed a ‘query session’ for ChatGPT regarding some questions about the diagnostic pathology and we added the answers in the figure 2-4. We hope that these features will be sufficient and explanatory. Thank you very much.

Reviewer n’2: b) Comparative Analysis: Given that only five publications met the review's criteria, a comparative analysis of these studies might offer a richer understanding. How do the findings of these papers align or differ, and what can we infer from these comparisons?

Answer n’5: Dear Reviewer n’2, thank you very much for this wonderful tip. We have added a table (Table 1) with the features of the 5 articles analyzed in our literature review.

Reviewer n’2: c) Addressing Limitations: The paper astutely points out several limitations, including data-related issues and "hallucination" phenomena. An exploration into potential solutions or workarounds for these limitations could be invaluable. For instance, are there new data sources or training techniques that could help overcome these challenges? d) Ethical Considerations: Beyond the technical and application-based discussions, it might be beneficial to touch upon the ethical implications of using AI like ChatGPT in a sensitive field like Pathological Anatomy. What are the potential risks, and how can they be mitigated?

Answer n’6: Dear reviewer n’2, thank you very much. We have added some sentences about the possible solutions of the actual issues.

Reviewer n’2: e) Future Roadmap: With the foundational knowledge laid out in the paper, readers might benefit from a section discussing the future roadmap or potential evolution of ChatGPT in this domain. What advancements or refinements can we expect in the coming years? As highlighted in the work of Schukow et al. it is imperative to outline future perspectives that the implementation of AI models will bring to the fore; first of all, it is important to consider how the use of AI methods applied to the writing of scientific articles will be managed, how to address the issue of consent and whether to modify the editorial lines of journals taking into account the use or otherwise of the chatbot. For the second thing, it will be very important to understand how ChatGPT can impact on the possible choice of specialization in Pathological Anatomy (increase or reduction); how and to what extent there will be a need for a critical review of the AI-generated content and whether or not the role of teacher/mentor can be delegated to ChatGPT.

Answer n’7: Thank you very much. Done.

Reviewer n’2: a) Limited Publications: From the outset, identifying 103 records and narrowing them down to merely five is quite limiting. The drastic reduction raises questions about the breadth of research available on the topic or the perhaps overly stringent inclusion criteria used by the authors. Such a limited pool can introduce biases and reduce the generalizability of the findings. b) Nature of Included Publications: Most publications were original articles, which, while valuable, represent a narrow slice of the scientific literature. Including only one case report, a letter to the editor, and a review must offer a diverse enough perspective on the potential applications and challenges of using ChatGPT in pathology.

Answer n’8: Dear Reviewer n’2, thank you very much. It’s a great and right observation. This field of application (pathology) is very poor explorated in literature and for this reason we have decided to call our review such as “preliminary”. Over time, we hope that this article will further stimulate AI-related research in fields related to Anatomic Pathology.

Reviewer n’2: c) Reliance on a 'Query Session': The decision to conduct a singular 'query session' to evaluate ChatGPT's competence may not offer a sufficiently rigorous assessment. Real-world scenarios often entail many complex cases, each requiring distinct considerations. A single session can hardly indicate the AI's capability across a diverse set of pathologies.

Answer n’9: Dear Reviewer n’2, thank you very much. We have performed others query session related to other types of diseases such as prostate adenocarcinoma and differential diagnosis between germ cell tumour of the ovary and serous high grade carcinoma of the ovary (Figure 5). Furthermore, we have performed another query sessions about immunohistochemical and molecular features of pleural mesothelioma and molecular characteristics of pediatric diffuse midline glioma. We hope that now there are many different topics related to human pathology.

Reviewer n’2: d) Absence of Comparative Analysis: The paper does not compare ChatGPT's responses to those of experienced pathologists or other AI systems. Such a comparative approach would offer a more explicit benchmark of the model's efficacy. e) Lack of Quantitative Metrics: The authors deemed the 'query session' results satisfactory. However, the criteria or metrics that led to this conclusion must be clarified. A more structured, perhaps quantitative, evaluation offered more weight to this assertion.

Answer n’10: Dear Reviewer n’2, thank you very much. Yes, surely a comparative analysis would be very interesting (for example between ChatGPT and expert pathologists) but in this paper we have performed a comparative analysis with standardized answers. Following the features of our check: “The ChatGPTs' responses were assessed individually by three reviewers, who checked each response to the standardized responses to evaluate the response's consistency. Based on their ranking, responses were classified as either "consistent" or "inconsistent." Throughout the evaluation process, there was strong agreement amongst the three reviewers, guaranteeing a consistent and unbiased assessment of the responses”.

Reviewer n’2: f) Over-reliance on AI Caution: While it is commendable that ChatGPT cautions against using its information without expert validation, leaning on this cautionary note as a safety net could be misleading. It is essential to assess the tool's reliability critically rather than depend on its in-built warnings.

Answer n’11: Thank you. We have reformulate these sentences.

Reviewer n’2: f) Over-reliance on AI Caution: While it is commendable that ChatGPT cautions against using its information without expert validation, leaning on this cautionary note as a safety net could be misleading. It is essential to assess the tool's reliability critically rather than depend on its in-built warnings.

Answer n’12: Thank you very much. Yes, the goal of our manuscript is to provide a “preliminary” vision of the use of ChatGPT in Diagnostic Pathology. Future researchs will be conducted to improve the quality of results.

Reviewer 3 Report

Comments and Suggestions for Authors

The authors state that although several clinical studies using ChatGPT have already been published in the literature, very little has yet been written about its potential application in human pathology. We conduct a systematic review following the Preferred Reporting Items for Systematic Reviews and Meta-Analyses (PRISMA) guidelines, using PubMed, Scopus and Web of Science (WoS) as databases, with the following keywords: ChatGPT OR Chat GPT, in combination with each of the following: pathology, diagnostic pathology, anatomic pathology. A total of 103 records were initially identified in the literature search, of which 19 were duplicates. After screening for eligibility and inclusion criteria, only 5 publications were ultimately included. The majority of publications were original articles (n = 2), followed by case reports (n = 1), letter to the editor (n = 1) and review (n = 1). According to the authors although the premises are exciting and ChatGPT is able to co-advise the pathologist in providing large amounts of scientific data for use in routine microscopic diagnostic practice, there are many limitations (such as data of training, amount of data available, ‘hallucination’ phenomena) that need to be addressed and resolved, with the caveat that an AI-driven system should always provide support and never a decision-making motive during the histopathological diagnostic process.

This is an interesting paper from a societal and scientifical point of view.

I have the following points to make the paper stronger:

Abstract: mention when the literature review has been conducted and when the chat with ChatGPT took place.

1.Introduction:

Add one or more research questions.

The authors state: In the first few months after its official launch, many papers were published in the purely informatic field, but, as the weeks went by, the medical-scientific field was also interested, with a particular interest in education, research and simulation of clinical pictures of patients, as well as applications in hygiene and public health, clinical medicine, oncology and surgery.” Give examples of studies form the medical-scientific field, and discuss them. This will allow the authors to embed their study in this field, and how it is related to ChatGPT.

3. Literature How the literature review has been conducted is well explained, though when the literature review has been conducted should be added.  But where are the results? Discuss the 5 studies.

Then add a new section in which you present the results of the chat with ChatGPT. Explain more in detail how you proceeded and when you did. Refer to Loos, E. F., Gröpler , J., & Goudeau, M. S. (2023). Using ChatGPT in education: Human reflection on ChatGPT’s self-reflection. Societies13(8), 196. https://doi.org/10.3390/soc13080196   who also chatted with ChatGPT and explained their procedure.

Conclusion:

Answer one or more research questions.

Add a Limitations section

Add a section discussing the implications for future studies of your study

Author Response

Reviewer n’3: Abstract: mention when the literature review has been conducted and when the chat with ChatGPT took place.

Answer n’1: Dear Reviewer n’2, first of all thank you very much for your precious suggestions. We will try to do our best to improve the quality of our manuscript. We have added the temporal features of the literature review and of the chat with ChatGPT in the abstract.

Reviewer n’3: 1.Introduction: Add one or more research questions. The authors state: In the first few months after its official launch, many papers were published in the purely informatic field, but, as the weeks went by, the medical-scientific field was also interested, with a particular interest in education, research and simulation of clinical pictures of patients, as well as applications in hygiene and public health, clinical medicine, oncology and surgery.” Give examples of studies form the medical-scientific field, and discuss them. This will allow the authors to embed their study in this field, and how it is related to ChatGPT.

Answer n’2: Dear Reviewer n’3, thank you very much for your suggestions. Done.

Reviewer n’3: 3. Literature How the literature review has been conducted is well explained, though when the literature review has been conducted should be added.  But where are the results? Discuss the 5 studies.

Answer n’3: Dear Reviewer, thank you very much. We added the temporal information of our review and we have added a table (table 1) with the features of the 5 studies included and that are discussed in the discussion.

Reviewer n’3: Then add a new section in which you present the results of the chat with ChatGPT. Explain more in detail how you proceeded and when you did. Refer to Loos, E. F., Gröpler , J., & Goudeau, M. S. (2023). Using ChatGPT in education: Human reflection on ChatGPT’s self-reflection. Societies, 13(8), 196. https://doi.org/10.3390/soc13080196   who also chatted with ChatGPT and explained their procedure.

Answer n’4: Thank you very much dear Reviewer n’3. We have added a section with details of the query sessions with ChatGPT.

Reviewer n’3: Conclusion: Answer one or more research questions.

Add a Limitations section

Add a section discussing the implications for future studies of your study

Answer n’5: Dear Reviewer n’3, thank you very much for this wonderful tip. Done.

Round 2

Reviewer 1 Report

Comments and Suggestions for Authors

After revision, the article became much better and, strange as it may seem, the authors convinced me that the article was worthy of publication. The publication of the article will make a significant contribution to the relevant field of science - the development of knowledge about preventing the use of ChatGPT in medicine. The use of artificial intelligence (and especially ChatGPT) in diagnosing patients is a crime against human health.

THE ARTICLE MUST BE PUBLISHED!!!

Reviewer 2 Report

Comments and Suggestions for Authors

The authors implemented all suggestion points mentioned in the previous revision round. The core gaps were solved and the overall quality of the paper improved. The reviewer's suggestion is to accept the paper in its present form. 

Reviewer 3 Report

Comments and Suggestions for Authors

In my opinion the paper can be published now.